# Mixed-state electron ptychography enables sub-angstrom resolution imaging with picometer precision at low dose

Zhen Chen [1], Michal Odstrcil [2,6], Yi Jiang[3], Yimo Han[1,7], Ming-Hui Chiu[4], Lain-Jong Li[4,8] & David A. Muller [1,5✉]

Both high resolution and high precision are required to quantitatively determine the atomic structure of complex nanostructured materials. However, for conventional imaging methods in scanning transmission electron microscopy (STEM), atomic resolution with picometer precision cannot usually be achieved for weakly-scattering samples or radiation-sensitive materials, such as 2D materials. Here, we demonstrate low-dose, sub-angstrom resolution imaging with picometer precision using mixed-state electron ptychography. We show that correctly accounting for the partial coherence of the electron beam is a prerequisite for high-quality structural reconstructions due to the intrinsic partial coherence of the electron beam. The mixed-state reconstruction gains importance especially when simultaneously pursuing high resolution, high precision and large field-of-view imaging. Compared with conventional atomic-resolution STEM imaging techniques, the mixed-state ptychographic approach simultaneously provides a four-times-faster acquisition, with double the information limit at the same dose, or up to a fifty-fold reduction in dose at the same resolution.

[1] School of Applied and Engineering Physics, Cornell University, Ithaca, NY 14853, USA. [2] Paul Scherrer Institut, 5232 Villigen PSI, Switzerland. [3] Advanced Photon Source, Argonne National Laboratory, Lemont, IL 60439, USA. [4] Physical Science and Engineering, King Abdullah University of Science and Technology, Thuwal 23955-6900, Saudi Arabia. [5] Kavli Institute at Cornell for Nanoscale Science, Ithaca, NY 14853, USA. [6]Present address: Carl Zeiss SMT, Carl-Zeiss-Straße 22, 73447 Oberkochen, Germany. [7]Present address: Department of Molecular Biology, Princeton University, Princeton, NJ 08544, USA. [8]Present address: Department of Electronic Engineering and Green Technology Research Center, Chang-Gung University, Taoyuan 333, Taiwan. ✉email: david.a.muller@cornell.edu

D etermining the local atomic arrangement of complex nanostructures can provide fundamental insights into the properties of materials[1,2]. Compared to traditional metals and semiconductors, newer materials systems as metal-organic frameworks and organic perovskites are more radiation sensitive[3–5], requiring more dose-efficient imaging techniques in order to allow high-resolution imaging with comparable level of detail. Solving the structure of biological macromolecules or small molecular at atomic level is even more challenging[6]. The main problem is that, as a consequence of Poisson statistics, the required illumination dose is inversely proportional to the square of the spatial resolution[7], and thus improving spatial resolution means quadratically higher doses. The increased dose may destroy the structure of the sample before sufficient image signal-to-noise is reached. The widely adopted atomic-resolution imaging methods in scanning transmission electron microscopy (STEM), such as annular dark-field (ADF) or coherent bright-field (cBF) imaging, are intrinsically dose inefficient as they use only a small fraction of the scattered electrons, being constructed via a simple integration of a limited portion of phase space. Therefore, conventional STEM imaging methods usually cannot achieve sub-angstrom resolution or even atomic-resolution for electron radiation-sensitive materials[8]. Meanwhile, high-precision measurement of local atomic positions is also fundamentally hindered by the poor signal-to-noise ratio of ADF images from electron-radiation sensitive or weakly scattering samples, such as monolayer 2D materials. Picometer precision via ADF imaging can only be achieved in electron-radiation-robust and strongly scattering bulk samples[9–11].

Electron ptychography, however, can potentially use the entire diffraction patterns either via a Wigner-distribution deconvolution (WDD)[12,13] or iterative algorithms[14,15] in a way that can account for the probe damping effect and extract the electrostatic potential of the sample. Electron ptychography has been demonstrated as a promising phase-contrast imaging technique with advantages such as high dose efficiency[13,16], high resolution[5,17,18], and high contrast[5,13]. In particular, ptychography has now surpassed the resolution of best physical lenses, reaching deep sub-angstrom resolution[5]. Simulations have suggested the possibility of extremely low-dose imaging by electron ptychography, in principle beyond that of all other electron imaging approaches to date including high-resolution TEM imaging widely used in Cryo-EM community—a potential outcome of considerable importance for the study of electron radiation-sensitive materials including biological macromolecules.

Imperfections of the STEM imaging system especially the partial coherence of the illumination probe reduces the image resolution and contrast of conventional STEM imaging methods[19]. Partial coherence also limits the performance of ptychography, although in a more indirect manner, impacting the signal-to-noise ratio of the reconstruction[5,20]. WDD-based electron ptychographic phase-contrast imaging has been demonstrated to outperform the commonly adopted phase-contrast conventional high-resolution TEM imaging by retaining resolution beyond the temporal incoherent limit[16,20]. However, the partial coherence in the probe can be modeled more explicitly in iterative ptychography algorithms by decomposing the probe wavefunction into a linear combination of pure states, i.e., a mixed quantum state[21]. As first demonstrated in coherent diffractive imaging[22] and X-ray ptychography[23], partial coherence of the probe can be well accounted for by introducing such state mixtures into the reconstruction method[21,24,25]. High coherent field-emission guns are widely used as the electron sources in modern electron microscopes and so electron ptychography usually assumes only a pure coherent state of the illumination probe[5,26,27]. This assumption is often sufficient when an in-focus illumination probe is adopted[5,27] or when a nanometer spatial resolution is targeted[26]. The effects of incoherence, which are inevitably present in electron ptychography, on the reconstruction quality remains underexplored. There are only a few proof-of-principle demonstrations of electron ptychography considering the partial coherence via approaches either Gaussian blind deconvolution[18] or modal decomposition[28,29], and while these suggest the promise of the approach, to date no sub-angstrom resolution reconstructions have been achieved, even on instruments capable of sub-angstrom resolution in conventional imaging modes.

Here, we demonstrate the capability of fast, sub-angstrom resolution and picometer-precision imaging with a large field-of-view (FOV) by mixed-state electron ptychography. We find that a complete description of the partial coherence of the electron wavefunction using a mixed quantum state is required to accomplish reconstructions with a sub-angstrom resolution, high contrast, high precision and a large FOV via defocused probe electron ptychography. We also demonstrate low-dose atomic-resolution ptychographic imaging using dose levels up to 50 times lower than conventional STEM imaging techniques.

## Results

**Experimental setup of electron ptychography.** Figure 1a shows a schematic workflow of the defocused electron ptychography. Instead of focusing the electron probe on the sample, the focal plane is set to a distance away from the sample. Because the illumination area is broadened, a larger scan step can be used given that there are sufficient overlaps between adjacent scan positions required by ptychographic reconstruction algorithms. The sample was raster scanned along two perpendicular directions as indicated in Fig. 1b. A high-dynamic-range electron microscope pixel-array detector (EMPAD)[30] used for diffraction pattern acquisition enables capture of both the shadow image[31] in the center disk formed by the large defocused probe and the much-weaker high scattering-angle dark-field signal, as illustrated by three simulated diffraction patterns in Fig. 1c. Similar setups with a defocused probe have been adopted previously[17,18,26], which has shown benefits for overcoming the slow readout speed of conventional CCD cameras and limited stability of the imaging systems. However, usually only a few dozen diffraction patterns were acquired, whereas fast pixel-array detectors like the EMPAD[30] enables the acquisition of more than 50,000 diffraction patterns per minute.

Conventional iterative ptychographic algorithms[14,15] assume a coherent illumination. The diffraction pattern is considered as the square of the amplitude of a single probe state multiplied by a sample transmission function within the multiplicative approximation (an extension of the strong phase approximation, allowing for both phase and amplitude variations)[12]. However, due to experimentally unavoidable partial coherence of the imaging system, single pure-state coherent probe illumination is never achieved in real experiments. To account for the partial coherence, the illumination is represented by several mutually incoherent probe modes instead of using only a single coherent probe mode[21]. Each mode is then propagated independently to the detector as the measured diffraction is an incoherent superposition of the contributions from all probe modes[21,28,32]. For practical implementations, we chose the modal decomposition approach[21,28] and the probe is expanded into several eigenmodes of the density matrix formed by a mixed state. The total intensity of all eigenmodes are normalized to the measured intensity of the diffraction patterns. A flowchart of the algorithms showing the basic principle is given in Supplementary Fig. 1.

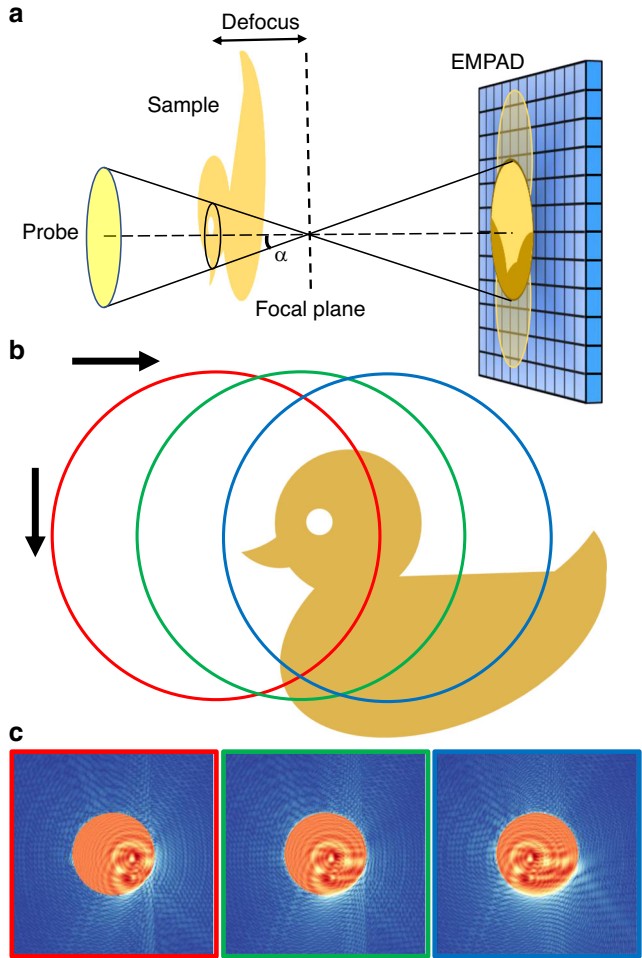

**Fig. 1 Schematic of defocused probe electron ptychography. a** Experimental setup. The focus of the electron probe is downstream from the sample at a distance defined by the defocus value. The diffraction pattern on the detector (EMPAD) shows a shadow image in the bright-field disk. **b** A diagram of the scan procedure. **c** Three diffraction patterns simulated when the probe is illuminated at the positions circled in **b**, which show the shadow images in the bright-field disk shifting accordingly with the probe position. The speckles in the dark-field region contain high frequency information.

Details of the reconstruction algorithms are summarized in the Methods section.

**Mixed-state ptychographic imaging to sub-angstrom resolution.** We performed a scanning diffraction experiment on a sample of a monolayer $WS_2$ with bilayer islands of $WS_2/MoSe_2$. The $MoSe_2$ and $WS_2$ layers have the same crystalline orientation, which is verified by the diffractogram of the ADF image, as shown in Supplementary Fig. 2. Owing to the 4% lattice mismatch between $WS_2$ and $MoSe_2$, the bilayer regions have a complex projected atomic structure with a continuous transition from a full Mo–Se (or W–S) bond length to intermediate misaligned projected distances (a structural model is given in Supplementary Fig. 2c). The resulting Moiré pattern serves as a good resolution test for our method.

Figure 2a shows a mixed-state ptychographic reconstruction of a dataset with an FOV of $30 \times 30$ nm$^2$ corresponding to $1500 \times 1500$ pixels. The selected sample region contains various structural features, such as a monolayer of $WS_2$ and both well-aligned and misaligned stacking bilayer $MoSe_2/WS_2$ regions. An enlarged view

of the bilayer region shown in Fig. 2b from the position marked on Fig. 2a shows the sliding structure of bilayer $WS_2/MoSe_2$. The Moiré-like pattern changes continuously from hexagonal rings in well-aligned stacking regions to stripe features in the misaligned regions (see the structural model in Supplementary Fig. 2c). Unambiguous sub-Ångstrom resolution can be illustrated in both Fourier space and real space. First, the isotropic information limit shown in the diffractogram of the reconstruction in Fig. 2d better than 1.4 Å$^{-1}$ and the diffraction spots circled on Fig. 2d correspond to a real-space distance of 0.69 Å. Second, the line profiles in Fig. 2e from atomic pairs marked on Fig. 2b demonstrate real-space peak separations down to 0.6 Å. In addition, the Fourier ring correlation (FRC) analysis[33], widely adopted for resolution estimation in cryo-electron microscopy and X-ray ptychography, also demonstrates spatial resolution down to 0.66 Å. This is shown in Supplementary Fig. 3, which is very close to the 0.69 Å resolution determined from the diffractogram in Fig. 2d. Figure 2c shows a conventional ADF image acquired with the same aperture size and a similar dose using an in-focused probe, limiting the resolution to worse than 1 Å. Compared with the ADF image in Fig. 2c, the ptychographic reconstruction in Fig. 2b demonstrates a significant and simultaneous improvement in contrast, signal-to-noise, and resolution.

What is more important is that the large FOV and sub-angstrom resolution ptychographic reconstruction are both achieved using a low-dose illumination. Typical doses used for atomic-resolution (1.5–2 Å) STEM ADF images of monolayer transition-metal dichalcogenides at 80 keV are ~$10^5$ e Å$^{-2}$ [34,35]. The dose for the dataset used in Fig. 2a is $1.6 \times 10^4$ e Å$^{-2}$. The ptychographic reconstruction has a 0.69 Å Abbe resolution, which doubles the resolution of ADF images achievable from the same imaging condition, ~1.37 Å (e.g., in Fig. 3). Furthermore, sub-angstrom resolution was reached for a dataset with doses down to $4.0 \times 10^3$ e Å$^{-2}$ (e.g., in Supplementary Fig. 4)—50 times lower than the dose for ADF images for the same resolution, with lower dose lattice images discussed below.

In addition, ptychography with a large defocused probe allows us to use a large scan step size of about 30% of the probe diameter[15,36], because the real-space pixel size of ptychographic reconstruction is determined by the maximum scattering angle of the diffraction pattern instead of scan step size in the conventional STEM ADF or BF images. This decouples the scan step size from the real-space sampling required by Nyquist–Shannon sampling theorem[37] for a certain resolution. This option enables a fast acquisition despite the frame-rate of current 2D array detectors being usually two to three orders slower than the acquisition rate of point detectors[30,38,39]. The dataset used for the ptychographic reconstruction shown in Supplementary Fig. 4a contains $64 \times 64$ diffraction patterns and the total acquisition time was 7.6 s. Assuming the same FOV and the typical acquisition conditions for ADF imaging—a step size of 0.2 Å with $1500 \times 1500$ pixels, a beam current of 10–40 pA, which requires 64–16 µs per pixel dwell time to achieve $10^5$ e Å$^{-2}$ dose[34,35]—the total acquisition time would be 144–36 s. Therefore, the high-quality ptychographic reconstruction with the same FOV is more than four times faster in acquisition while still using one order of magnitude lower dose compared with conventional ADF imaging.

Further tests of the practical experimental conditions show that much relaxed real-space overlap constraints, defined as $(1 - r/D)$[36], where $r$ is the scan step size and $D$ is the diameter of probe, is required by mixed-state ptychography. As shown in Supplementary Fig. 5, mixed-state ptychography with only two probe modes provides a stable high-quality reconstruction for a much larger scan step size up to 5.08 Å, corresponding to only 72% probe overlap, whereas artifact free reconstruction by single-state ptychography can only be achieved with a very small step size, 0.85 Å, corresponding to a 95% probe overlap. For larger scan step

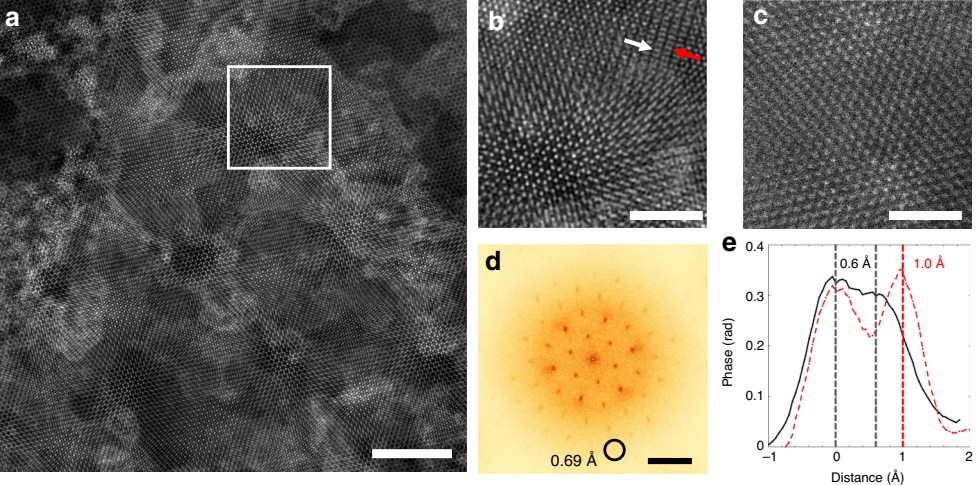

**Fig. 2 Mixed-state electron ptychographic reconstruction of a bilayer MoSe$_2$/WS$_2$ sample. a** Reconstruction from 128 × 128 diffraction patterns with a scan step size of 2.36 Å, field of view of 30 × 30 nm$^2$, and reconstructed pixel size of 0.2 Å, which is close to the pixel size (0.19 Å) in ADF image in **c**. Scale bar is 5 nm. **b** Enlarged images from the regions marked by the white rectangular on **a** and showing the bilayer structure. The white and red arrows point to the atomic pairs for the intensity profiles in **e**. **c** Conventional annular dark-field (ADF) image from a similar bilayer area acquired in a focused probe condition using the same aperture size and a similar dose as that for ptychography in **a**. Scale bars in **b** and **c** are 2 nm. **d** Diffractogram of the whole reconstruction in **a**, the circulated diffraction spot corresponds to a 0.69 Å distance in real space. Scale bar is 0.7 Å$^{-1}$. **e** Line profiles across two atomic pairs pointed by the white and red arrows on **b** showing atomic column distances of 0.6 and 1.0 Å, respectively.

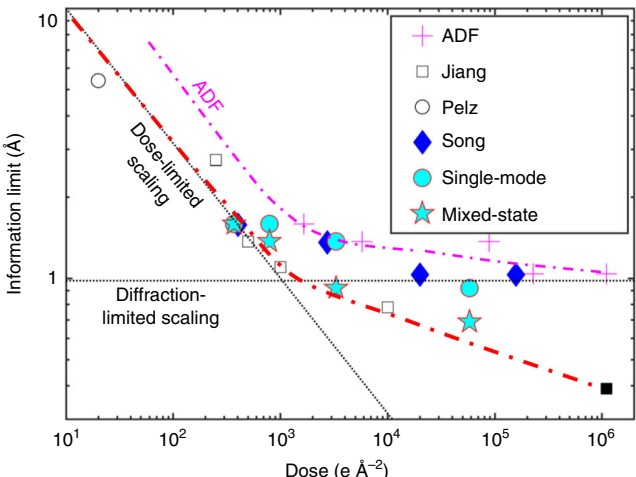

**Fig. 3 Summary of dose dependence of information limit from ADF and electron ptychography.** The opened markers are electron ptychographic results using simulated data from references Jiang et al. and Pelz et al. The filled markers are experimental results by electron ptychography from Song et al. and this manuscript. The single-mode and mixed-state results are from Fig. 4 in the main text. Resolution by dose-limit, inversely proportional to square root of dose (linear in log-log plot), and by aperture limit are guided by dashed lines. Information limit for ADF (plus makers) is measured from low-angle ADF images of monolayer WS$_2$ synthesized from in-focused 4D-STEM.

sizes, shown in Supplementary Fig. 5b, c, the single-state reconstructions fail to converge to the correct structure and generate artificial periodicities of the sample. At large step sizes, the errors in modeling of the probe partial coherence by the single-state reconstruction accumulate, leading to a solution that is not able to describe the measured data well and therefore, in combination with the weaker constrains due to larger scanning step, it leads to nonphysical reconstructions. Even in the case of a large probe overlap where the single-state reconstruction works, 95%, the mixed-state reconstruction has about two times better

resolution and enhanced contrast than that from the single-state (Supplementary Fig. 5). We also find that four eigenmodes are sufficient to capture the incoherence properties of our probe, and additional modes do not lead to improvement of the reconstruction quality. A comparison of reconstructions with different numbers of modes is presented in Supplementary Fig. 6. Therefore, under practical conditions of partially coherent illumination, mixed-state ptychography significantly relaxes the requirement of the real-space constraint, i.e., scan step, and makes large FOV atomic-resolution imaging feasible.

**Low-dose atomic-resolution imaging.** Low-dose electron ptychography was first suggested via the Wigner-distribution deconvolution (WDD) approach[13], but only $10^4$ e Å$^{-2}$ or higher dose for atomic-resolution imaging has been realized experimentally by this approach[16]. In principle, the low-dose performance should be comparable for WDD and iterative ptychography algorithms[5], but the high sampling density required and currently slow detector readouts make working at low dose more challenging for WDD[40]. For iterative electron ptychography, the dose-limit has been explored previously via simulations[5,18,41,42] and some experimental attempts[40]. Simulations usually show one to two orders lower dose than what has been realized by experiments[5]. This discrepancy may be caused by experimental limitations, such as sample stability, limited beam stability, and partial coherence, which were neglected in simulations. Recent work using a highly coherent defocused probe and single-state electron ptychography has demonstrated a lattice resolution image from a monolayer MoS$_2$ using low-dose illumination[40]. However, only resolution worse than 1 Å was achieved and partial coherence of the illumination was not considered in the ptychographic reconstruction. To explore the impact of partial coherence on the practical limit of the low-dose imaging, we show the dose dependence of ptychographic reconstructions using experimental datasets and find benefits for both resolution and required dose after accounting for partial coherence.

Figure 4 shows a series of reconstructions using datasets acquired from monolayer WS$_2$ by varying the illumination dose via changing the beam current. For a dose higher than 3300 e Å$^{-2}$

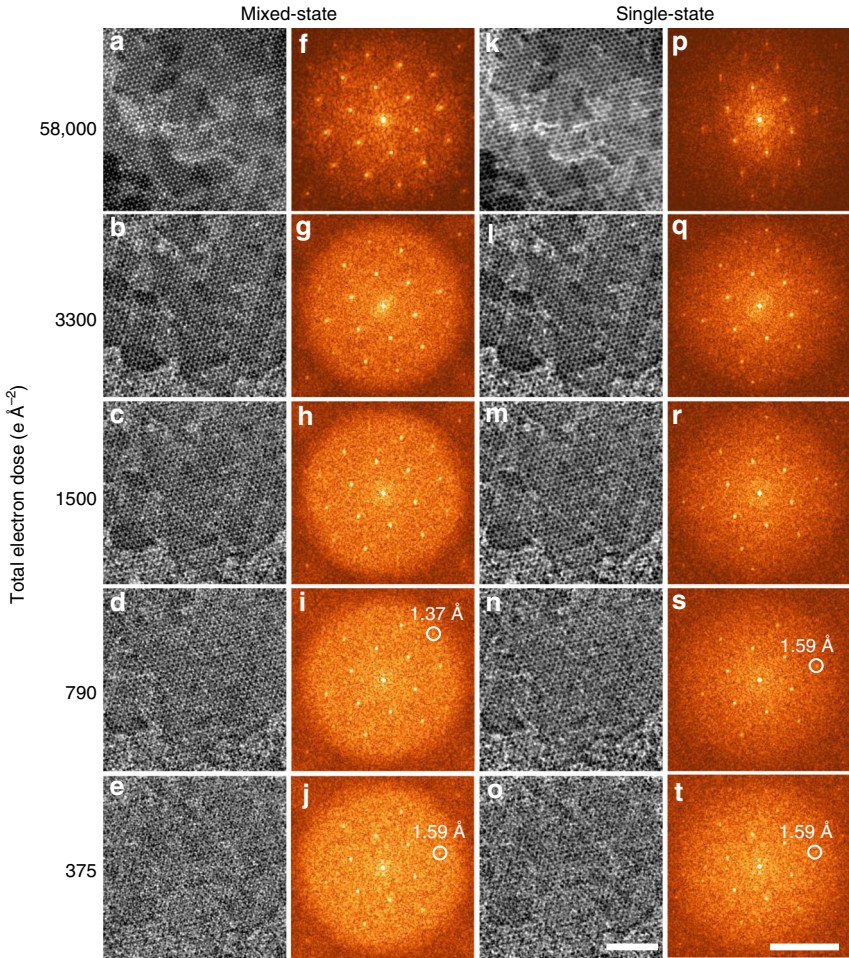

**Fig. 4 Ptychographic reconstructions of a monolayer WS$_2$ sample in different illumination dose conditions. a–e** Reconstructions by mixed-state ptychography using doses 58,000, 3300, 1500, 790, and 375 e Å$^{-2}$, respectively. **k–o** Reconstructions by single-state ptychography. **f–j**, **p–t** Corresponding diffractograms of the reconstructions. Scan step size is 0.85 Å for doses from 790 to 58000 e Å$^{-2}$. For the dose of 375 e Å$^{-2}$, the dataset is selected in an interval of two from the dataset with a dose of 1500 e Å$^{-2}$ and the scan step size is 1.69 Å. Three probe modes were used in mixed-state ptychography. The real-space distance of the representative diffraction spots is labeled on **i**, **j** and **s**, **t**. The non-uniform contrast in the reconstructed images comes from the polymer residual during the sample preparation. Scale bars for real-space images in **a–e** and **k–o** are 3 nm, and for diffractograms in **f–j** and **p–t** are 0.7 Å$^{-1}$.

(Fig. 4a, b), both the W and S sublattice can be resolved unambiguously and the isotropic information transfer is higher than 1.1 Å$^{-1}$, corresponding to a real-space resolution better than 0.9 Å as shown in Fig. 4f, g. For doses of 1500 and 790 e Å$^{-2}$ shown in Fig. 4c, d, the reconstruction quality is slightly reduced but both sublattices are still resolvable and the information transfer is higher than 0.73 Å$^{-1}$, corresponding to a real-space resolution of 1.37 Å. Even for a dose of 375 e Å$^{-2}$, shown in Fig. 4e, the lattice can still be recognized, and the information transfer is up to ~0.63 Å$^{-1}$, corresponding to a real-space resolution of 1.59 Å. As mentioned above, the dose commonly used for atomic-resolution (~1.5–2.0 Å) ADF images is about $10^5$ e Å$^{-2}$ [34,35]. Therefore, the minimum dose in the ptychographic reconstruction with atomic-resolution demonstrated here is more than two orders smaller.

We also compared the reconstructions from mixed-state and single-state ptychography in the low-dose conditions. As demonstrated in Fig. 4a–e, k–o, both the resolution and contrast of the reconstructions from mixed-state ptychography are enhanced compared with single-state ptychography using the same dose. The mixed-state ptychographic reconstruction has a better resolution at half the dose than that for single-state reconstruction; see for example, Figs. 4m and 4d. Similarly, the mixed-state reconstructions

in Fig. 4 show better quality and higher information limit compared with single-state reconstructions at similar doses reported recently[40]. A summary of the dose-resolution dependence including the results in Fig. 4 and previous reported ptychographic results[5,40,41] is given in Fig. 3. Some general trends and scaling behaviors can be noted, even though the samples and experimental conditions for all the data are not identical, which may slightly affect the absolute resolution value at a certain dose. In the low dose-limit where performance is limited by counting statistics, the resolution is well-aligned along a family of dose-limit lines, $k/\sqrt{N}$. $N$ is the dose, and $k$ is a constant that depends on the imaging method and the scattering power of the diagnosed samples[43]. Compared with conventional ADF images, for a certain information limit, the dose required for the mixed-state ptychographic reconstructions is 10–50 times lower. At higher doses, the resolution is limited by the largest angle of information transfer. Ptychography is limited by the cut-off angle of the diffraction patterns used in the reconstruction and can demonstrate a much better resolution limit than ADF imaging, which is limited by the probe-forming aperture.

The negligible readout noise (1/40 e$^-$) and single-electron sensitivity of detector used[30] is critical for approaching the limits of the low-dose ptychographic reconstruction. As shown in

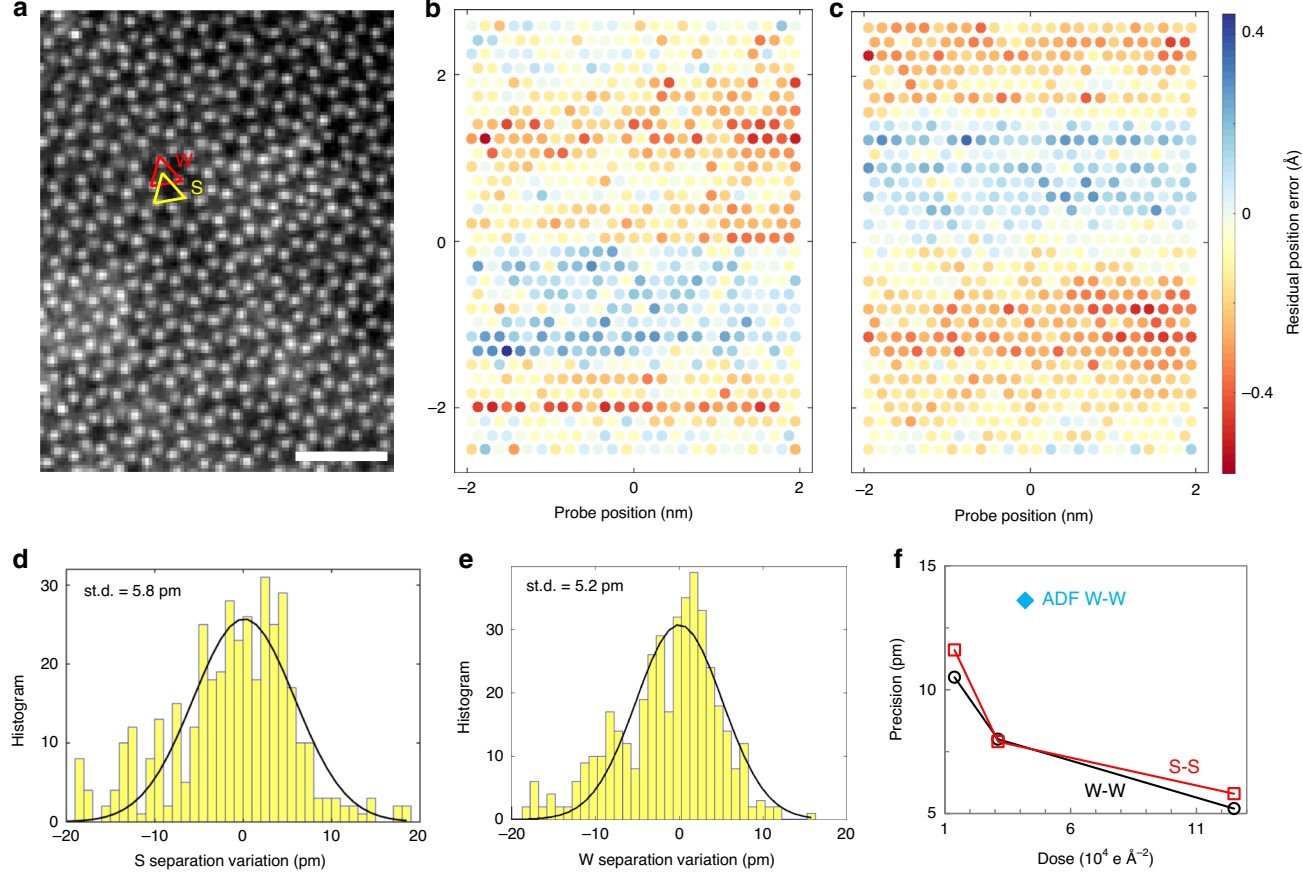

**Fig. 5 Measurement precision of atomic distance. a** A ptychographic reconstruction from a monolayer $WS_2$ sample using illumination dose of $1.25 \times 10^5$ e Å$^{-2}$. W–W and S–S sublattices are labeled with a red and yellow triangle respectively. Scale bar is 1 nm. **b, c** Residual probe position errors along horizontal and vertical direction estimated by ptychography. The original probe position is shown as circles and the random position error is displayed by the false color. **d**, **e**. Statistical distribution of S–S and W–W atomic separations. **f** Precision of W–W and S–S distance in dependence on the illumination dose. Precision of W–W distance from a conventional ADF image with a dose $4.2 \times 10^4$ e Å$^{-2}$ denoted by a diamond symbol on **f**.

Supplementary Fig. 7a–d, a single diffraction pattern with dose smaller than 3300 e Å$^{-2}$ contains only a few electrons per pixel in the center disk and a very large fraction of pixels with fewer than single electron in dark-field region. Despite the very low dose, which provides only ~500 electrons per scan position for a dose of 790 e Å$^{-2}$, the averaged diffraction patterns shown in Supplementary Fig. 7e–h still retain sharp edges of the diffraction disks. This demonstrates that the interference between different diffraction paths has been well encoded in these sparse-signal diffraction patterns. The lattice fringes in Fig. 4d reveal that the sample phase information contained in the low-dose diffraction patterns have been retrieved via electron ptychography.

**High-precision atomic position measurements.** One of the advantages of ptychography is the ability to correct for positioning errors[44–46] by maximization of mutual consistency between the adjacent regions. These methods can serve as an independent estimation of the position measurements, which can in principle correct for sample drifts and improve the precision of atomic position measurement.

We have selected an area of monolayer $WS_2$ ($4.0 \times 5.2$ nm$^2$) shown in Fig. 5a. The contrast from light sulfur elements in the ptychographic reconstruction is very strong, because the contrast of the phase image is roughly linearly proportional to $Z$, the atomic number[47], whereas the contrast in ADF is close to a $Z^2$ dependence[8,48]. Therefore, atomic positions of sulfur atoms can

be readily obtained with high precision even in the vicinity of heavy W atoms. In conventional ADF images, the sulfur atoms are shadowed by the strong scattering from W atoms, which hinders the estimation of the sulfur positions. The recorded EMPAD dataset shows sample drift during the data acquisition. However, the relative probe position errors can be very well estimated via the gradient-based probe position refinement algorithm[49]. At first, the global geometry errors, such as scan step size scaling, axis rotation or skewness, are fitted and corrected via an affine transformation between the nominal probe positions and estimated probe positions. Owing to the raster position scanning, a constant velocity drift of the sample will result in a skewness of up to a few degrees for long acquisition datasets. When the geometry model is refined, the residual position errors are estimated. Figure 5b, c shows the residual position errors along the horizontal (fast scan) and vertical (slow scan) direction, respectively. The standard deviation (st.d.) of the residual errors along both directions shown in Fig. 5b, c is ~0.16 Å with a maximum error of ~0.6 Å. The atomic column positions were estimated using a 2D Gaussian function fit of the reconstructed phase. The positions were used to calculate the nearest neighbor W–W and S–S atomic distances. The distributions of the S–S and W–W distances are plotted in Fig. 5d, e. The st.d. of the S–S distance is 5.8 pm and the st.d. of W–W distance is 5.2 pm, which is a precision measurement of the atomic positions as the monolayer $WS_2$ region shown in Fig. 5a has

negligible structural distortions. The intensity variations in Fig. 5a are from residual polymer residue, and likely degrade the precision of the bond-length measurements. Therefore, the reported precision of the ptychography method should be viewed as an upper bound.

For transition-metal dichalcogenides imaged by 80 keV electrons, the dose must be kept below ~$10^6$ e Å$^{-2}$ to avoid significant structural damage[50]. An even lower dose beneath $10^4$ e Å$^{-2}$ is required to avoid the formation of sulfur point vacancies or structural alterations around deficient regions[51]. Therefore, it is important to see the performance of ptychography for estimating atomic positions at low dose. Figure 5f shows the dose-dependent precision estimation of W–W and S–S distances. Precisions of about 10 pm for both W–W and S–S distances can still be achieved using doses as low as $10^4$ e Å$^{-2}$. However, from a conventional ADF image, the precision of W–W atomic distance with a single fast scan (6 μs per pixel chosen to match dose and pixel sampling) is only 13.6 pm as shown in Supplementary Fig. 8, which is about twice as large as that from ptychography with a similar dose. Using a more stable imaging system, multiple scans and drift correction algorithms, precision for both ADF imaging and ptychography can be further improved[10,11]. As detector speeds increase, the multiple scan strategy becomes more practical for ptychography. Denoising and deconvolution algorithms can also help with peak location[41,52]. However, sulfur atoms in the ADF image are not visible due to their much-weaker scattering, shown in Supplementary Fig. 8a. Although sulfur atoms can be seen by using a lower collection angle for ADF[35], the precision of S atomic positions determined from low-dose ADF image is much worse than that from W. Therefore, the low-dose and high contrast capabilities of ptychography provide a picometer-precision technique for atomic position determination of single atoms including light sulfur atoms in 2D materials.

## Discussion

We have demonstrated that mixed-state electron ptychography provides simultaneously improved imaging capabilities, including high resolution, large FOV, low dose, high contrast/signal-to-noise ratio, and high precision. A mixed-state model is beneficial for ptychographic reconstructions using the data from current electron microscopes, as it is able to account for many of the different factors that result in decoherence-like effects in measured diffraction patterns[20,21,42,53], such as the finite electron source size, chromatic aberration, sample vibration, and fast instabilities of the image-forming system. With further improved coherence of the electron source, such as a cold field-emission gun, a larger scan step sizes with a reduced overlap ratio could be achieved[54]. Loss of speckle contrast in diffraction, which has similar effects as partial spatial coherence of the probe, can be introduced due to a finite detector pixel size and limit the largest applicable probe size[20,55]. Therefore, with further improvement of the source coherence and the area of the detectors, increased scan step sizes can be utilized, which further enlarges the FOV given the same number of scanning positions. Furthermore, requirements of the illumination stability in ptychography can be relaxed by use of multiple scans with shared object information[56] or other probe relaxation extension[49,57]. These approaches can largely overcome the long-term stability limitations of the current scanning systems, which makes even micron length scale with sub-angstrom resolution imaging feasible.

Ptychography requires a forward model for the interaction of the beam with the sample. One limitation of the current mixed-state ptychographic imaging is that it can only be applied in relatively thin samples because it uses a generalized strong phase approximation that neglects the effects of beam propagation. The

generalized phase grating approximation for the interaction of the incident probe with a projected object function can be written as, $\psi_{exit}(\mathbf{r}_i, \mathbf{r}) = \psi_{in}(\mathbf{r} - \mathbf{r}_i)O(\mathbf{r})$, where $\psi_{exit}(\mathbf{r}_i, \mathbf{r})$ is the electron wavefunction passing through the sample, $\psi_{in}(\mathbf{r} - \mathbf{r}_i)$ is the incident electron wavefunction centered at position $\mathbf{r}_i$, with $\mathbf{r}_i$ and $\mathbf{r}$ being 2D coordinates. The complex object function, $O(\mathbf{r})$ is a generalized strong phase object, $O(\mathbf{r}) = A(\mathbf{r})\exp(i\sigma V(\mathbf{r}))$, where $A(\mathbf{r})$ is the amplitude, $\sigma$ is the interaction constant depending on the electron energy and $V(\mathbf{r})$ is the projected electrostatic potential of the sample. The amplitude term is included to allow for a weak absorption effect, e.g., scattering outside the detector and should be close to unity if the sample is thin[58]. Failures of the model for practical samples might be suspected if the amplitude either deviates by more than 10% from unity or resembles the phase term such that phase-amplitude mixing has likely occurred[59].

If the probe shape changes significantly during propagation within the sample, then the probe–sample interaction cannot be well described in a single plane and a full multi-slice calculation may need to be considered. Both probe free-space propagation and scattering by the sample can change the probe shape. The thickness limit $T$ due to the propagation effect can be expressed as[60], $T = 1.3\lambda/\theta_{max}^2$, where $\theta_{max}$ is the maximum scattering angle of the diffraction pattern and $\lambda$ is the wavelength of electrons. For a typical scattering angle targeting a resolution better than 0.5 Å, $\theta_{max} = 20$ mrad at 300 keV, the thickness limit is ~6.4 nm, which is within the achievable thickness for many samples. For heavy elements, a single atom can induce a large phase shift and a strong amplitude damping to the electron wavefunction, and the probe shape can be changed significantly by only a few atoms. Therefore, a much more rigorous thickness limit must be adapted for samples containing high atomic number elements[61]. Recent attempts to solve the multiple scattering problem in thick samples include multi-slice ptychography[60,62,63] and scattering matrix phase retrieval[64]. Although the robustness and convergence must be further improved to achieve practical applications in thick samples[63,65], mixed-state ptychography could be extended to include multiple scattering[49].

Data processing speed is another limiting factor for applications of ptychography. However, with graphics processing unit (GPU) acceleration, the reconstruction of the large FOV image shown in Fig. 2a only takes less than one hour on a typical GPU card. The processing time largely scales linearly with number of diffraction patterns, therefore, fewer patterns with the defocused probe setup can significantly accelerate the reconstruction.

The flexibility and robustness of mixed-state electron ptychography enable many potential applications. Large FOV high resolution imaging is critical for uncovering both the overall morphology and the local atomic arrangement in complex nanostructures[66]. High contrast and high precision coupled with low-dose imaging can be used to measure the atomic scale dynamics of light elements, such as Li, O, or S in lithium battery materials. Fast single-pass acquisition with scan drift correction opens the door to in situ phase-contrast STEM imaging of dynamical processes during heating, cooling, or even chemical reaction. Use of a large-illuminated-area probe enables a larger scan step size and fewer diffraction patterns for a given FOV, which reduces the computational effort during data reconstruction and analysis and accelerates data processing of ptychographic reconstruction. Rapid data processing is critical for live imaging and 3D structural reconstruction, such as ptychographic tomography[67,68]. Our demonstration of low-dose imaging at atomic-resolution is within the allowable dose for many beam-sensitive materials[69]. Further dose reduction could be potentially realized by improvements of the reconstruction algorithm and experimental setup such as the averaging of multiple images from

structurally identical particles that is commonly used in single-particle Cryo-EM. The high dose efficiency of mixed-state ptychography may also be helpful for reconstructing biological molecules using cryo-electron microscopy, potentially reducing the number of particles needed in an averaging class to achieve a desired resolution[41]. With the anticipated next generation pixel-array detectors that will be even larger and faster, the probe size can be further increased and thus a larger scan step size can be chosen, which will further enlarge the FOV and increase the acquisition speed. Furthermore, mixed-state electron ptychography, besides the retrieval of the probe mixture, can also be adapted to retrieve mixed quantum states within a sample[21,32], which could expand the reach of quantum-state tomography by using a matter wave.

## Methods

**Experimental method**. The scanning diffraction experiments were carried out using an electron microscope pixel-array detector (EMPAD)[30] installed on a probe aberration corrected Thermo Scientific™ Titan Themis electron microscope. The EMPAD has $128 \times 128$ pixels, a readout speed of 0.86 ms per frame, and 1,000,000:1 electron linear response. All the datasets were acquired using a probe at 80 keV beam energy, 21.4 mrad probe-forming semi-angle, and ~55 nm defocus value. The exposure time was 1 ms per frame. The beam current varied from 0.09 to 14.3 pA via defocusing a monochromator. The coherence of the electron probe increases slightly when the beam current reduces but the change is not significant (<3%) as the beam current used (0.1–14 pA) is always much lower than the coherent current of the source (~50 pA). The large $30 \times 30$ nm$^2$ FOV image in Fig. 2 is reconstructed from a dataset with a scan step size of 2.36 Å containing $128 \times 128$ diffraction patterns, which were selected in an interval of two from a larger dataset with $256 \times 256$ diffraction patterns. A reconstruction of a four-times down-sampled dataset with a 4.72 Å scan step size is shown in Supplementary Fig. 4. The imaged sample contains MoSe$_2$ islands on a large area of mono-layer WS$_2$. The relative orientation of MoSe$_2$ and WS$_2$ is identical, as we have verified using the Fourier transform of the ADF image shown in Supplementary Fig. 2.

**Mixed-state ptychography**. We adapted the generalized maximum-likelihood ptychography method[49,70] initially developed for X-ray ptychography. This method solves the phase retrieval problem by preconditioned gradient descent-based optimization. Optimization of amplitude likelihood function[49,70] provides more robust convergence than direction optimization of Poisson likelihood. Multiple optimization directions, such as probe and object updates, probe position displacements, or wavefront variation can be carried out jointly in a consistent way. In combination with a neural-networks-inspired mini-batch optimization scheme[49], our approach enables a compromise between convergence speed and noise robustness, and thus it significantly improves the usability of ptychography for low-dose imaging.

The mixed-state description is implemented using the modal decomposition approach[21]. Minor time variations of the illumination probe due to the instability of the electron optics or small sample height variation are accounted for by using an illumination wavefront correction method[49], which is a computationally faster and more memory-efficient approximation of the orthogonal probe relaxation (OPR) approach[57]. The OPR method describes small variations of the illumination probe by a linear decomposition into a set of several orthogonal mutually coherent modes. The workflow of the algorithm is schematically shown in Supplementary Fig. 1 and more details are described as a ptychography toolkit in ref.[71].

We have observed that the reconstruction quality of the mixed-state ptychography algorithm is not sensitive to the initial guess of the illumination probe and the number of the mixed modes for datasets with a sufficient probe overlap, although a good initial guess may accelerate the convergence. On the other hand, the single probe mode ptychography requires a good initial probe guess to provide stable convergence for our experimental datasets. This seems counterintuitive but it is not surprising. Because mixed-state ptychography can account for the nonnegligible partial coherence of the probe and provide a more accurate reciprocal model, whereas single-mode ptychography is not sufficient for modeling the probe incoherence and its convergence can become unstable if the initial probe deviates from the real probe significantly.

**Fourier ring correlation**. For Fourier ring correlation (FRC)[33,72], we used two phase images reconstructed from two separate datasets from the same scan region, which serves as two independent measurements. Practically, two datasets were selected from one single dataset in every two diffractions at each dimension but with different starting points. After ptychographic reconstruction, a global linear phase term due to the inherent ambiguities of ptychography[73] is removed by fitting as a 2D linear function. Two phase images are aligned using the sub-pixel precision cross-correlation algorithm[74]. Before FRC analysis, the edges of the phase images were Apodized to avoid the artifacts introduced from the boundary discontinuities[33]. The resolution is estimated by using the 1-bit threshold[33].

## Data availability

A small (200 Mb) data subset is available from PARADIM, a National Science Foundation Materials Innovation Platform [https://doi.org/10.34863/G4WA-0J57][75]. Full datasets are available from the corresponding author (david.a.muller@cornell.edu) on request.

## Code availability

The codes developed at Cornell University is published on GitHub, muller-group-cornell [https://github.com/muller-group-cornell]. The ptychography reconstruction toolkit, PtychoShelves developed at Paul Scherrer Institut, Switzerland, is available on the website [https://www.psi.ch/en/sls/csaxs].

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

## Acknowledgements

Z.C. and D.A.M. are supported by the PARADIM Materials Innovation Platform program in-house program by NSF Grant DMR-1539918. Y.H. is supported by the NSF MRSEC program (DMR-1429155). This work made use of the Cornell Center for Materials Research facility supported by NSF grant DMR-1719875. We thank Tsai Esther Hsiao Rho and Manuel Guizar-Sicairos for useful discussions.

## Author contributions

Experiments and data analysis were performed by Z.C. under the supervision of D.A.M. The main ptychographic algorithms were implemented by M.O. for X-ray ptychography with the adaption to electron ptychography by Z.C. and Y.J. Sample preparation was done by Y.H. from thin films synthesized by M.C. and L.J. Z.C. wrote the manuscript with revisions from M.O. and D.A.M. All authors discussed the results and implications throughout the investigation, and all authors have given approval to the final version of the manuscript.

## Competing interests

Cornell University has licensed the EMPAD hardware to Thermo Scientific.
