## [Peer Review File · Nature Communications]

Reviewers' comments:

Reviewer #1 (Remarks to the Author):

This manuscript reports the application of mixed-state electron ptychography to image 2D transition-metal dichalcogenides (TMDs) with sub-angstrom resolution and picometer precision at low electron dose. Using a bilayer MoSe₂/WS₂ and a monolayer WS₂ sample, the authors demonstrated that, compared with conventional STEM imaging techniques, mixed-state ptychography provides a four-time-faster acquisition with double the information limit at the same dose. Although ptychography was proposed in 1969, the modern version of ptychography using iterative algorithms was experimentally demonstrated with x-rays in 2017 [PRL 98, 034801 (2007)], which was based on the coherent diffractive imaging (CDI) experiment in 1999 [Nature 400, 342–344 (1999)]. Ptychography has since been actively pursued in the x-ray field due the rapid development of coherent x-ray sources and pixel array detectors. As a result, in terms of the methodology development, x-ray ptychography has thus far been ahead of electron ptychography. But, with the availability of fast pixel-array detectors for electrons, the situation can be changed as electron ptychography has already shown some unique advantages over conventional S/TEM methods. In my opinion, this work represents an important experiment in this emerging field. The data analysis is very solid and the manuscript is well written. Therefore, I recommend its publication in Nature Communications, provided the following points are fully addressed.

1. In this work, the authors have used 2D TMDs as samples, where the multiple scattering effects are negligible. But for many important samples, the multiple scattering effects may be significant and higher energy electrons such as 200 keV are preferred. I suggest the authors to briefly discuss whether the conclusions present in the abstract are applicable to general samples with higher energy electrons.
2. In the first paragraph in page 4, the authors stated that “Compared to x-rays, electron sources have a higher brightness and longitudinal coherence (typically, $\Delta E/E < 10^{-5}$) and so electron ptychography usually assumes only a pure coherent state of the illumination probe”. This statement is incorrect and needs to be fixed. As x-rays are bosons and electron are fermions, x-rays can have much higher brightness than electrons. For example, the Europe x-ray free electron laser can reach x-ray brilliance to 10^{26} photon/s/0.1%bw/mm²/mrad². Also, some inelastic x-ray scattering beamlines at the state-of-the-art synchrotron radiation facilities can have much higher energy resolution than 10^{-5} .
3. In the 2nd paragraph in page 9, the authors stated that “Typical doses used for atomic resolution (1.5~2 Å) STEM ADF images of monolayer TMDs at 80 keV are $\sim 10^5$ e/Å².” Also, in Fig. 5, the authors show a 2D precision of 5 - 6 pm with a dose of 1.25×10^5 e/Å². But the authors did not compare their results with the true state-of-the-art results on ADF STEM. For example, a recent article [arXiv:1901.00633 (2019)] shows that scanning atomic electron tomography using ADF STEM can determine the 3D atomic positions and crystal defects in Re-doped MoS₂ with a 3D precision down to 4 pm. The total dose of the 13 2D projection images is 4.1×10^5 e/Å² (i.e. 3.15×10^4 e/Å² per projection). I suggest

the authors to briefly discuss their results and scanning atomic electron tomography of monolayer TMDs.

4. In terms of the methodology development, I believe that there is much to learn between the x-ray and electron CDI/ptychography fields. I thus suggest the authors to cite a recent review on x-ray CDI / ptychography [Science 348, 530-535 (2015)], which will be of interest to the electron ptychography community.

Reviewer #2 (Remarks to the Author):

This manuscript presents results and a discussion of ptychographic imaging in the STEM, using the multiple overlap of probes approach, including partial coherence of the STEM probe, and with data acquired with the new EMPAD detector. The main thrust is that the approach can lead to better resolution/precision for a given electron dose, or meet the resolution/precision of currently well utilized techniques using a lower dose. This is really very nice work, and it is potentially important for the wider materials community to know that in the field of electron microscopy work is being done to reduce the dose requirements for atomic resolution imaging, since the technique has gained notoriety for high radiation dose.

On the other hand, while the manuscript is mostly quite well well-written, it appears written mostly with a specialized audience in mind. More specifically, the introduction is well written and general, but later sections quickly become specialized, containing details that a more general audience would find difficult to appreciate. Because of this, I would invite the authors to revise their manuscript to make it more accessible to a general audience.

Points where revision is needed:

- * Along the lines of making the manuscript more accessible to a wider audience, Figs 3 and 4 look similar, and both involve changing the electron dose.
- * Also along the lines of making the manuscript more accessible, there is very little discussion of what is actually imaged beyond "The Moiré-like pattern changes continuously from hexagonal rings in well-aligned stacking regions to stripe features in the misaligned regions." The authors could expand on this in order to put the images in context for a non-specialist audience. Could pull Suppl Fig 1c into main text, for example.
- * Suppl Fig 4 would be interesting for the general audience. Perhaps pull into main text?
- * There is no discussion of the algorithm or offline processing in the main text. But it is a vital part of the scheme so that such a discussion should be included, i.e., an outline of the algorithm, typical processing times, some discussion of reconstruction errors for single vs mixed state, etc.
- * Why does the lattice look wrong in some of the single state reconstructions? (e.g. Fig 3c and Suppl Fig 4a) The authors briefly mention "false minima," but it is surprising that the peaks in the Fourier transform appear in the wrong place. I would have expected that the phase in Fourier space would be wrong, but not so much the amplitude.
- * It would certainly be fair to include a discussion of some of the disadvantages of ptychographic approaches. For example, compared to ptychographic approaches, one great benefit of ADF for experimenters is that it requires no offline processing. Another benefit is less data.
- * The abstract reads "We show that correctly disentangling the mixed-state electron wave function is a prerequisite for high-quality structural reconstructions due to the intrinsic partial coherence of the electron beam." This sounds like quantum entanglement, but the authors really mean something else, and this could be confusing/misleading for those outside the field.
- * It reads "Electron ptychography, however, can potentially use the entire diffraction patterns either via a Wigner-distribution deconvolution (WDD)^{12,13} or iterative algorithms^{14,15} in a way that can account for the probe damping effect and extract the electrostatic potential of the

sample." But ptychography only takes care of the imaging process, and then we must assume a relationship to the electrostatic potential (unless we do multislice...), just as in conventional imaging, so this statement appears misleading.

* It reads "decomposing the probe wave function into a linear combination of pure states, i.e., mixed quantum states." But it is a single mixed state.

* Similarly later on it reads, "using multiple mixed quantum states" but again just a single mixed state.

* "A mixed-state model of the electron probe ...is able to account many of the different factors ... such as the finite electron source size, chromatic aberration, point spread function of the detector, sample vibration, and fast instabilities of the image-forming system" Should the detector PSF be included in this list? Finite source size (sum of incoherent probes) does reduce the interference between different diffraction paths, but the sharpness of the diffraction discs remains the same.

* In Fig 4 the caption reads "monolayer WS2" but from the images the sample appears to have multiple layers.

* I assume that Fig 5 d-f assumes a perfect WS2 sample? What is influence of multiple layers/adatoms/contamination on the sample?

* Fig 5 suggest to make b and c the same size as a.

* It reads "the used detector..." :) Perhaps "detector used..."

Reviewer #3 (Remarks to the Author):

The authors demonstrate that a ptychography reconstruction algorithm taking into account partial coherence of the probe significantly outperforms a single-mode reconstruction. This is shown for experimental data of a thin-film heterostructure with Moiré structures arising from a lattice mismatch and single-layer regions, where the absolute precision of lengths can be gauged. Partial coherence is modeled through a decomposition of the probe into mutually incoherent eigenmodes, which as far as I understand are refined along with the sample.

The probably most significant outcome is that for a given resolution, the dose can be massively reduced with respect to both HAADF-STEM (less surprising) and conventional ptychography. The latter is a striking result and it can safely be assumed that it will have a significant impact on the electron ptychography community, which so far was stuck with single-mode reconstructions. While unfortunately even the lowest doses studied are 1-2 orders of magnitude away from what would be required for biological systems, where radiation damage is the limiting factor to state-of-the-art methods, I understand that this is beyond the scope of this paper.

The results achieved by the reconstruction method are remarkable, and publication in Nature Communication is absolutely warranted. The manuscript is easy to follow, picks a good selection of references and comparison to previous work, and includes extensive and helpful supplementary information.

There is, however, one omission that should be addressed: while the outcome of the reconstruction algorithm is nicely showcased, the algorithm itself is explained only very briefly in the methods section, which I think is insufficient given that it is the major finding. For the most

part, it seems like a partial-coherence extension to Odstroil et al. 2018 - it would be very helpful to have a more detailed and general description of how the simultaneous refinement of the mutually incoherent modes is implemented, and what numerical problems arise specifically from this. Maybe even graphical depiction (flow chart-like) of how the iterative algorithm does its work would be possible. As is, the paper leaves a bit of a "black magic" impression in that regard.

Some other points:

To study the different doses, it is stated that the current has been varied. It has to be discussed how this was done, and what impact it has on the coherence properties of the beam. Given that the brightness (current/phase space volume) along the beam line is a conserved quantity, coherence and current are hard to disentangle, e.g. increasing the probe demagnification by exciting the C1 lens (concomitantly decreasing the current) should improve coherence etc.... of course there are effective sources of decoherence like sample vibrations, chromatic aberrations etc - but this needs to be discussed.

In a similar vein, is there a conjecture about how the shapes of the incoherent modes arise?

Especially the third mode with its sixfold symmetry looks interesting.

Is there an intuitive reason why many-mode reconstruction is less sensitive to the initial mode guess? It feels counterintuitive, given the introduction of another degree of freedom to a strongly non-convex problem.

It is stated that an FRC analysis has been performed to estimate the resolution of the shown reconstruction. This requires more explanation. In single-particle analysis, an FRC between two single-particle reconstructions derived from data half-sets serves as an estimator for the spectral SNR. However, as no single-particle analysis is performed on identical particles in the images, it is not clear to me between which data sets FRC is calculated, and how a resolution criterion can be defined.

The color map of figure 2d is inverted with respect to all others. If this is intentional, please state, why.

I find it unsatisfactory that the software containing the key algorithm is only accessible on request instead of being immediately available on a public repository where it can be used and reviewed by other researchers, ideally under a standard open source license. What is the rationale behind this decision?

With these points addressed, I support publication of the paper in Nature Communications.

The edits of the texts in the main manuscript have been labeled in red. In response to the comments of the referees, we have made several changes to the manuscript. Main changes are listed as follows with details response to reviewers' comments listed after each comment.

1. The original Fig. 3 is moved to Supplementary Fig. 4. The sub-section "sparse scan of defocused electron ptychography" relating to the original Fig. 3 has been shortened and merged to the previous sub-section.
2. We have moved original Supplementary Fig. 5 to the main text as Fig. 4.
3. We have also moved the original main text Fig. 4 to Supplementary Fig. 4 and moved the original Supplementary Fig. 4 to Supplementary Fig. 5.
4. We have added more details about the numerical methods in the Methods section.

Response to reviewer #1:

This manuscript reports the application of mixed-state electron ptychography to image 2D transition-metal dichalcogenides (TMDs) with sub-angstrom resolution and picometer precision at low electron dose. Using a bilayer MoSe₂/WS₂ and a monolayer WS₂ sample, the authors demonstrated that, compared with conventional STEM imaging techniques, mixed-state ptychography provides a four-time-faster acquisition with double the information limit at the same dose. Although ptychography was proposed in 1969, the modern version of ptychography using iterative algorithms was experimentally demonstrated with x-rays in 2017 [PRL 98, 034801 (2007)], which was based on the coherent diffractive imaging (CDI) experiment in 1999 [Nature 400, 342–344 (1999)]. Ptychography has since been actively pursued in the x-ray field due the rapid development of coherent x-ray sources and pixel array detectors. As a result, in terms of the methodology development, x-ray ptychography has thus far been ahead of electron ptychography. But, with the availability of fast pixel-array detectors for electrons, the situation can be changed as electron ptychography has already shown some unique advantages over conventional S/TEM methods. In my opinion, this work represents an important experiment in this emerging field. The data analysis is very solid and the manuscript is well written. Therefore, I recommend its publication in Nature Communications, provided the following points are fully addressed.

We greatly appreciate the referee's positive comments and recommendations for publication in Nature Communications.

1. In this work, the authors have used 2D TMDs as samples, where the multiple scattering effects are negligible. But for many important samples, the multiple scattering effects may be significant and higher energy electrons such as 200 keV are preferred. I suggest the authors to briefly discuss whether the conclusions present in the abstract are applicable to general samples with higher energy electrons.

We appreciate the referee's suggestion and agree with the comment that multiple scattering effects are present and important in general thicker samples. All the discussions in this manuscript are based on the thin sample approximation, or more explicitly, multiplicative approximation, an extension of the strong phase approximation. As tested for X-ray ptychography in Ref [60], the thickness limit, T , is estimated to be, $T = \frac{1.3\lambda}{\theta_{max}^2} \approx 6.4$ nm, for 300 keV electrons and maximum scattering angle, $\theta_{max} = 20$ mrad, which is within the achievable thickness for many samples. For samples thicker than T imaged at 300 keV, the reconstruction quality of the conventional ptychography will drop progressively in dependence on the

scanning overlap. If the violation of the multiplicative approximation is the major limitation, modified ptychography algorithms such as multi-slice ptychography are required. The limitation of this has been added in the discussion part on page 19, lines 19-23, page 20, and page 21, lines 1-2:

Ptychography requires a forward model for the interaction of the beam with the sample. One limitation of the current mixed-state ptychographic imaging is that it can only be applied in relatively thin samples because it uses a generalized strong phase approximation that neglects the effects of beam propagation. The generalized phase grating approximation for the interaction of the incident probe with a projected object function can be written as, $\psi_{exit}(\vec{r}_i, \vec{r}) = \psi_{in}(\vec{r} - \vec{r}_i)O(\vec{r})$, where $\psi_{exit}(\vec{r}_i, \vec{r})$ is the electron wavefunction passing through the sample, $\psi_{in}(\vec{r} - \vec{r}_i)$ is the incident electron wavefunction centered at position \vec{r}_i , with \vec{r}_i and \vec{r} being 2D coordinates. The complex object function, $O(\vec{r})$, is a generalized strong phase object, $O(\vec{r}) = A(\vec{r})\exp(i\sigma V(\vec{r}))$, where $A(\vec{r})$ is the amplitude, σ is the interaction constant depending on the electron energy and $V(\vec{r})$ is the projected electrostatic potential of the sample. The amplitude term is included to allow for a weak absorption effect, e.g., scattering outside the detector and should be close to unity if the sample is thin⁵⁸. Failures of the model for practical samples might be suspected if the amplitude either deviates by more than 10% from unity or resembles the phase term such that phase-amplitude mixing has likely occurred⁵⁹.

If the probe shape changes significantly during propagation within the sample, then the probe-sample interaction cannot be well described in a single plane and a full multislice calculation may need to be considered. Both probe free-space propagation and scattering by the sample can change the probe shape. The thickness limit T due to the propagation effect can be expressed as⁶⁰, $T = 1.3\lambda/\theta_{max}^2$, where θ_{max} is the maximum scattering angle of the diffraction pattern and λ is the wavelength of electrons. For a typical scattering angle targeting a resolution better than 0.5 Å, $\theta_{max} = 20$ mrad at 300 keV, the thickness limit is ~ 6.4 nm, which is within the achievable thickness for many samples. For heavy elements, a single atom can induce a large phase shift and a strong amplitude damping to the electron wavefunction, and the probe shape can be changed significantly by only a few atoms. Therefore, a much more rigorous thickness limit must be adapted for samples containing high atomic number elements⁵⁷. Recent attempts to solve the multiple scattering problem in thick samples include multi-slice ptychography⁵⁸⁻⁶⁰ and scattering matrix phase retrieval⁶¹. Although the robustness and convergence must be further improved to achieve practical applications in thick samples^{60,62}, mixed-state ptychography could be extended to include multiple scattering⁴⁹.

2. In the first paragraph in page 4, the authors stated that “Compared to x-rays, electron sources have a higher brightness and longitudinal coherence (typically, $\Delta E/E < 10^{-5}$) and so electron ptychography usually assumes only a pure coherent state of the illumination probe”. This statement is incorrect and needs to be fixed. As x-rays are bosons and electron are fermions, x-rays can have much higher brightness than electrons. For example, the Europe x-ray free electron laser can reach x-ray brilliance to 1026 photon/s/0.1%bw/mm2/mrad2. Also, some inelastic x-ray scattering beamlines at the state-of-the-art synchrotron radiation facilities can have much higher energy resolution than 10^{-5} .

We realized that both the x-rays and electron sources are improving rapidly nowadays. Furthermore, direct comparison between x-rays and electrons in terms of coherence and brightness is complicated, which is beyond the scope of the current manuscript. Therefore, we avoid a direct comparison between xrays and electrons and changed the statement in page 4, lines 8-9, to,

High coherent field emission guns are widely used as the electron sources in modern electron microscopes.

3. In the 2nd paragraph in page 9, the authors stated that “Typical doses used for atomic resolution (1.5~2 Å) STEM ADF images of monolayer TMDs at 80 keV are $\sim 10^5$ e/Å².” Also, in Fig. 5, the authors show a 2D precision of 5 - 6 pm with a dose of 1.25×10^5 e/Å². But the authors did not compare their results with the true state-of-the-art results on ADF STEM. For example, a recent article [arXiv:1901.00633 (2019)] shows that scanning atomic electron tomography using ADF STEM can determine the 3D atomic positions and crystal defects in Re-doped MoS₂ with a 3D precision down to 4 pm. The total dose of the 13 2D projection images is 4.1×10^5 e/Å² (i.e. 3.15×10^4 e/Å² per projection). I suggest the authors to briefly discuss their results and scanning atomic electron tomography of monolayer TMDs.

We thank the reviewer for bringing our attention to the new work. We agree that using advanced image registration schemes with fast multiple scans, sophisticated deconvolution and denoising procedures, the precision can be further improved. However, for identical acquisition conditions, ptychography has higher dose efficiency than ADF and the capacity for self-consistent drift correction. Furthermore, the position of light elements, such as sulfur cannot be accurately determined from low-dose ADF images. We have added the discussions on page 18, lines 13-16:

Using a more stable imaging system, multiple scans and drift correction algorithms, precision for both ADF imaging and ptychography can be further improved^{10,11}. As detector speeds increase, the multiple scan strategy becomes more practical for ptychography. Denoising and deconvolution algorithms can also help with peak location^{41,52}.

Ref. [52] has been added:

52. Tian, X. *et al.* Correlating 3D atomic defects and electronic properties of 2D materials with picometer precision. *arXiv:1901.00633v3* (2019).

4. In terms of the methodology development, I believe that there is much to learn between the x-ray and electron CDI/ptychography fields. I thus suggest the authors to cite a recent review on x-ray CDI / ptychography [Science 348, 530-535 (2015)], which will be of interest to the electron ptychography community.

We agree that much can be learned between the x-ray and electron CDI/ptychography fields. We have added the suggested reference and a more recent review about x-ray ptychography as Refs [23] and [24] in the introduction part in page 4, line 7.

23. Miao, J., Ishikawa, T., Robinson, I. K. & Murnane, M. M. Beyond crystallography: Diffractive imaging using coherent x-ray light sources. *Science* 348, 530-535 (2015).

24. Pfeiffer, F. X-ray ptychography. *Nat. Photonics* 12, 9-17 (2017).

Response to reviewer #2:

This manuscript presents results and a discussion of ptychographic imaging in the STEM, using the multiple overlap of probes approach, including partial coherence of the STEM probe, and with data acquired with the new EMPAD detector. The main thrust is that the approach can lead to better resolution/precision for

a given electron dose, or meet the resolution/precision of currently well utilized techniques using a lower dose. This is really very nice work, and it is potentially important for the wider materials community to know that in the field of electron microscopy work is being done to reduce the dose requirements for atomic resolution imaging, since the technique has gained notoriety for high radiation dose.

On the other hand, while the manuscript is mostly quite well well-written, it appears written mostly with a specialized audience in mind. More specifically, the introduction is well written and general, but later sections quickly become specialized, containing details that a more general audience would find difficult to appreciate. Because of this, I would invite the authors to revise their manuscript to make it more accessible to a general audience.

We thank the reviewer's constructive suggestions. To make the manuscript more easily understand for general audience, we have reformulated the manuscript thoroughly. The main changes are:

1. The original Fig. 3 is moved to Supplementary Fig. 4. The sub-section "sparse scan of defocused electron ptychography" relating to the original Fig. 3 has also been shortened and merged to the previous sub-section.
2. We have moved original Supplementary Fig. 5 to the main text as Fig. 4.
3. We have also moved the original main text Fig. 4 to Supplementary Fig. 4 and moved the original Supplementary Fig. 4 to Supplementary Fig. 5.
4. All the corresponding texts have been modified.

Points where revision is needed:

** Along the lines of making the manuscript more accesible to a wider audience, Figs 3 and 4 look similar, and both involve changing the electron dose.*

We have moved Fig 3 to Supplementary Fig. 5 and changed the discussions accordingly.

** Also along the lines of making the manuscript more accesible, there is very little discussion of what is actually imaged beyond "The Moiré-like pattern changes continuously from hexagonal rings in well-aligned stacking regions to stripe features in the misaligned regions." The authors could expand on this in order to put the images in context for a non-specialist audience. Could pull Suppl Fig 1c into main text, for example.*

The period of the Moiré pattern is very large, ~10 nm. Many atoms must be included in order to show the continuously structural change in a structural model, which requires a picture with many pixels. It could not be fit into the Fig. 1. We refer to Supplementary Fig. 1c in case a general audience is interested, on page 7, line 16 and page 8, line 2.

** Suppl Fig 4 would be interesting for the general audience. Perhaps pull into main text?*

We have put Supplementary Fig. 4 as a main Fig. 4 and the original Fig. 4 is moved to Supplementary Fig 4.

** There is no discussion of the algorithm or offline processing in the main text. But it is a vital part of the scheme so that such a discussion should be included, i.e., an outline of the algorithm, typical processing times, some discussion of reconstruction errors for single vs mixed state, etc.*

We have added a link to code in the code-availability section, an outline of the algorithm (and references to more details) on page 7, lines 3-5.

For practical implementations, we chose the modal decomposition approach^{24,28} and the probe is expanded into several eigenmodes of the density matrix formed by a mixed state. The total intensity of all eigenmodes are normalized to the measured intensity of the diffraction patterns.

We have also added the data processing speed in the discussion part on page 21, lines 3-7.

Data processing speed is another limiting factor for applications of ptychography. However, with graphics processing unit (GPU) acceleration, the reconstruction of the large FOV image shown in Fig. 2a only takes less than one hour on a typical GPU card. The processing time largely scales linearly with number of diffraction patterns, therefore, fewer patterns with the defocused probe setup can significantly accelerate the reconstruction.

** Why does the lattice look wrong in some of the single state reconstructions? (e.g. Fig 3c and Suppl Fig 4a) The authors briefly mention "false minima," but it is surprising that the peaks in the Fourier transform appear in the wrong place. I would have expected that the phase in Fourier space would be wrong, but not so much the amplitude.*

The single state reconstructions do not consider the nonnegligible partial coherence of the used probe. In Fig. 3c and Suppl. Fig 4a, a very large scan step size and thus a less constrained condition was used. The minimal solution found by the single-state reconstruction is not able to describe the measured data well and therefore, in combination with the weaker constraints due to larger scanning step, it leads to nonphysical reconstructions. In addition, ptychography is different from the traditional iterative phase retrieval methods, such as Gerchberg and Saxton (Optik **35**, 237 (1972)), which usually directly measure a lower resolution amplitude of the object image and only reconstruct the missing phase. Therefore, it is much easier to get the amplitude correctly reconstructed in the traditional phase retrieval methods. But ptychography measures the intensity of diffraction patterns and does not use any real-space image as an input. A similar constraint is still required, which fulfills via the overlap between neighboring probe positions. If the overlap is not enough, the structure of the object can be completely wrong. The wrong lattice of single state reconstructions indicates that more overlap or stronger constraints are required for single state reconstruction in the case of partial coherent illumination. We have added some statements in the main text on page 10, line 17-22, and the Figure captions of Supplementary Fig. 4 and 5.

For larger scan step sizes, shown in Supplementary Fig. 4b-c, the single-state reconstructions fail to converge to the correct structure and generate artificial periodicities of the sample. Because the inadequate modeling of the probe partial coherence by the single-state reconstruction leads to a solution that is not able to describe the measured data well and therefore, in combination with the weaker constraints due to larger scanning step, it leads to nonphysical reconstructions.

(In figure caption of Supplementary Fig. 4)

The single-mode reconstruction can result in an incorrect lattice structure if the overlap is not sufficient, such as in **b** and **c**.

(In figure caption of Supplementary Fig. 5)

The single-mode reconstruction in **a** shows an incorrect lattice structure due to over-simplified reconstruction model without counting for the nonnegligible partial coherence of the probe.

** It would certainly be fair to include a discussion of some of the DISadvantages of ptychographic approaches. For example, compared to ptychographic approaches, one great benefit of ADF for experimenters is that it requires no offline processing. Another benefit is less data.*

We thank the reviewer's constructive suggestions. We have added paragraphs stating the main limitations of ptychography on page 19, lines 19-23, page 20, and page 21, lines 1-7:

Ptychography requires a forward model for the interaction of the beam with the sample. One limitation of the current mixed-state ptychographic imaging is that it can only be applied in relatively thin samples because it uses a generalized strong phase approximation that neglects the effects of beam propagation. The generalized phase grating approximation for the interaction of the incident probe with a projected object function can be written as, $\psi_{exit}(\vec{r}_i, \vec{r}) = \psi_{in}(\vec{r} - \vec{r}_i)O(\vec{r})$, where $\psi_{exit}(\vec{r}_i, \vec{r})$ is the electron wavefunction passing through the sample, $\psi_{in}(\vec{r} - \vec{r}_i)$ is the incident electron wavefunction centered at position \vec{r}_i , with \vec{r}_i and \vec{r} being 2D coordinates. The complex object function, $O(\vec{r})$, is a generalized strong phase object, $O(\vec{r}) = A(\vec{r})\exp(i\sigma V(\vec{r}))$, where $A(\vec{r})$ is the amplitude, σ is the interaction constant depending on the electron energy and $V(\vec{r})$ is the projected electrostatic potential of the sample. The amplitude term is included to allow for a weak absorption effect, e.g., scattering outside the detector and should be close to unity if the sample is thin⁵⁸. Failures of the model for practical samples might be suspected if the amplitude either deviates by more than 10% from unity or resembles the phase term such that phase-amplitude mixing has likely occurred⁵⁹.

If the probe shape changes significantly during propagation within the sample, then the probe-sample interaction cannot be well described in a single plane and a full multislice calculation may need to be considered. Both probe free-space propagation and scattering by the sample can change the probe shape. The thickness limit T due to the propagation effect can be expressed as⁶⁰, $T = 1.3\lambda/\theta_{max}^2$, where θ_{max} is the maximum scattering angle of the diffraction pattern and λ is the wavelength of electrons. For a typical scattering angle targeting a resolution better than 0.5 Å, $\theta_{max} = 20$ mrad at 300 keV, the thickness limit is ~ 6.4 nm, which is within the achievable thickness for many samples. For heavy elements, a single atom can induce a large phase shift and a strong amplitude damping to the electron wavefunction, and the probe shape can be changed significantly by only a few atoms. Therefore, a much more rigorous thickness limit must be adapted for samples containing high atomic number elements⁵⁷. Recent attempts to solve the multiple scattering problem in thick samples include multi-slice ptychography⁵⁸⁻⁶⁰ and scattering matrix phase retrieval⁶¹. Although the robustness and convergence must be further improved to achieve practical applications in thick samples^{60,62}, mixed-state ptychography could be extended to include multiple scattering⁴⁹.

Data processing speed is another limiting factor for applications of ptychography. However, with graphics processing unit (GPU) acceleration, the reconstruction of the large FOV image shown in Fig. 2a only takes less than one hour on a typical GPU card. The processing time largely scales linearly with number of

diffraction patterns, therefore, fewer patterns with the defocused probe setup can significantly accelerate the reconstruction.

** The abstract reads "We show that correctly disentangling the mixed-state electron wave function is a prerequisite for high- quality structural reconstructions due to the intrinsic partial coherence of the electron beam." This sounds like quantum entanglement, but the authors really mean something else, and this could be confusing/misleading for those outside the field.*

We have changed the sentence as,

We show that correctly accounting for the partial coherence of the electron beam is a prerequisite for high-quality structural reconstructions.

** It reads "Electron ptychography, however, can potentially use the entire diffraction patterns either via a Wigner-distribution deconvolution (WDD)^{12,13} or iterative algorithms^{14,15} in a way that can account for the probe damping effect and extract the electrostatic potential of the sample." But ptychography only takes care of the imaging process, and then we must assume a relationship to the electrostatic potential (unless we do multislice...), just as in conventional imaging, so this statement appears misleading.*

We describe in more detail the relationship between exit wave phase and the electrostatic potential used in the reconstruction. In thin samples, this is essentially just a single step of the multislice algorithm. For thicker samples, multi-slice ptychography can in principle retrieve the electrostatic potential of the sample at different planes with resolution along beam axis limited by the depth of focus. To make this clearer, we have written explicitly the main approximation that ptychography uses and the limitations for thick samples in the discussion part on page 19, lines 19-23, page 20, and page 21, lines 1-2:

Ptychography requires a forward model for the interaction of the beam with the sample. One limitation of the current mixed-state ptychographic imaging is that it can only be applied in relatively thin samples because it uses a generalized strong phase approximation that neglects the effects of beam propagation. The generalized phase grating approximation for the interaction of the incident probe with a projected object function can be written as, $\psi_{exit}(\vec{r}_i, \vec{r}) = \psi_{in}(\vec{r} - \vec{r}_i)O(\vec{r})$, where $\psi_{exit}(\vec{r}_i, \vec{r})$ is the electron wavefunction passing through the sample, $\psi_{in}(\vec{r} - \vec{r}_i)$ is the incident electron wavefunction centered at position \vec{r}_i , with \vec{r}_i and \vec{r} being 2D coordinates. The complex object function, $O(\vec{r})$, is a generalized strong phase object, $O(\vec{r}) = A(\vec{r})\exp(i\sigma V(\vec{r}))$, where $A(\vec{r})$ is the amplitude, σ is the interaction constant depending on the electron energy and $V(\vec{r})$ is the projected electrostatic potential of the sample. The amplitude term is included to allow for a weak absorption effect, i.e., scattering outside the detector and should be close to unity if the sample is thin⁵⁸. Failures of the model for practical samples might be suspected if the amplitude either deviates by more than 10% from unity or resembles the phase term such that phase-amplitude mixing has likely occurred⁵⁹.

If the probe shape changes significantly during propagation within the sample, then the probe-sample interaction cannot be well described in a single plane and a full multislice calculation may need to be considered. Both probe free-space propagation and scattering by the sample can change the probe shape. The thickness limit T due to the propagation effect can be expressed as⁶⁰, $T = 1.3\lambda/\theta_{max}^2$, where θ_{max} is the maximum scattering angle of the diffraction pattern and λ is the wavelength of electrons. For a typical scattering angle targeting a resolution better than 0.5 Å, $\theta_{max} = 20$ mrad at 300 keV, the thickness limit is ~ 6.4 nm, which is within the achievable thickness for many samples. For heavy elements, a single atom

can induce a large phase shift and a strong amplitude damping to the electron wavefunction, and the probe shape can be changed significantly by only a few atoms. Therefore, a much more rigorous thickness limit must be adapted for samples containing high atomic number elements⁵⁷. Recent attempts to solve the multiple scattering problem in thick samples include multi-slice ptychography⁵⁸⁻⁶⁰ and scattering matrix phase retrieval⁶¹. Although the robustness and convergence must be further improved to achieve practical applications in thick samples^{60,62}, mixed-state ptychography could be extended to include multiple scattering⁴⁹.

** It reads "decomposing the probe wave function into a linear combination of pure states, i.e., mixed quantum states." But it is a single mixed state.*

We have changed all the 'mixed quantum states' to 'a mixed quantum state'.

** Similarly later on it reads, "using multiple mixed quantum states" but again just a single mixed state.*

We have corrected to 'using one mixed quantum state'.

** "A mixed-state model of the electron probe ...is able to account many of the different factors ... such as the finite electron source size, chromatic aberration, point spread function of the detector, sample vibration, and fast instabilities of the image-forming system" Should the detector PSF be included in this list? Finite source size (sum of incoherent probes) does reduce the interference between different diffraction paths, but the sharpness of the diffraction discs remains the same.*

The detector PSF degrades the sharpness of the diffraction disc and reduces the coherence, details can be found in Ref. [23] and [24]. We have already included those references. The incoherence from the finite source size comes from the incoherent sum of diffractions from different spatial positions. Such incoherent summation results in reduced contrast of the coherent speckles due to the different phase change of the probe wave function induced by the sample. Therefore, finite source size can be naturally covered within mixed-state model. A good discussion has been given in Ref. [24].

** In Fig 4 the caption reads "monolayer WS₂" but from the images the sample appears to have multiple layers.*

Fig 3 (original Fig. 4) shows monolayer WS₂ regions. The weak contrast change from area-to-area are from the polymer residues not multiple layers. This can be distinguished from the nonregular contrast and lack of edge structures. We added a statement to clarify this in the figure caption of Fig. 3 (original Fig. 4).

The non-uniform contrast in the reconstructed images comes from the polymer residual during the sample preparation.

** I assume that Fig 5 d-f assumes a perfect WS₂ sample? What is influence of multiple layers/adatoms/contamination on the sample?*

In order to estimate the precision of the ptychography technique, we use a monolayer WS₂ region, with no vacancies, shown in Fig. 5, in which all W-W and S-S distances are assumed to be the same. However,

there are unavoidable distortions on the projected structure from mobile adatoms or polymer contaminants. The variations shown in Fig 5 d-f include all these effects, with the polymer residue perhaps the most likely limiting factor for this measurement. To distinguish contributions from different sources to the structural variations, a more controlled experiment is needed which is not the focus of this work. Therefore, we only set an upper limit of the precision of the ptychography method.

We have added two sentences on page 18, lines 1-3:

The intensity variations in Fig 5a are from residual polymer residue, and likely degrade the precision of the bond-length measurements. Therefore, the reported precision of the ptychography method should be viewed as an upper bound.

** Fig 5 suggest to make b and c the same size as a.*

We have adjusted the figure size to make Fig. 5a the same size as b and c.

** It reads "the used detector..." :) Perhaps "detector used..."*

We have changed to 'detector used'.

Response to reviewer #3:

The authors demonstrate that a ptychography reconstruction algorithm taking into account partial coherence of the probe significantly outperforms a single-mode reconstruction. This is shown for experimental data of a thin-film heterostructure with Moiré structures arising from a lattice mismatch and single-layer regions, where the absolute precision of lengths can be gauged. Partial coherence is modeled through a decomposition of the probe into mutually incoherent eigenmodes, which as far as I understand are refined along with the sample.

The probably most significant outcome is that for a given resolution, the dose can be massively reduced with respect to both HAADF-STEM (less surprising) and conventional ptychography. The latter is a striking result and it can safely be assumed that it will have a significant impact on the electron ptychography community, which so far was stuck with single-mode reconstructions. While unfortunately even the lowest doses studied are 1-2 orders of magnitude away from what would be required for biological systems, where radiation damage is the limiting factor to state-of-the-art methods, I understand that this is beyond the scope of this paper.

We thank the reviewer for their comments. The doses required for an image depend on both the scattering power of the sample and the resolution. Lower dose at similar resolution might be achievable in other samples with stronger scattering. The dose required for atomic resolution in biological materials may be still too high for a single reconstruction. But averaging reconstructions from several separate but structural identical particles, like in single particle Cryo-EM, is still applicable. And the higher dose efficiency means

that fewer particles are required to achieve a desired resolution. We have added one sentence on page 21, lines 22-23 and page 22, lines 1-2:

The high dose efficiency of mixed-state ptychography may also be helpful for reconstructing biological molecules using cryo-electron microscopy, potentially reducing the number of particles needed in an averaging class to achieve a desired resolution⁴¹.

The results achieved by the reconstruction method are remarkable, and publication in Nature Communication is absolutely warranted. The manuscript is easy to follow, picks a good selection of references and comparison to previous work, and includes extensive and helpful supplementary information.

There is, however, one omission that should be addressed: while the outcome of the reconstruction algorithm is nicely showcased, the algorithm itself is explained only very briefly in the methods section, which I think is insufficient given that it is the major finding. For the most part, it seems like a partial-coherence extension to Odstroil et al. 2018 - it would be very helpful to have a more detailed and general description of how the simultaneous refinement of the mutually incoherent modes is implemented, and what numerical problems arise specifically from this. Maybe even graphical depiction (flowchart-like) of how the iterative algorithm does its work would be possible. As is, the paper leaves a bit of a "black magic" impression in that regard.

We have added descriptions of the implementation methods on page 7, lines 4-7, in Method section on page 23 and also a flowchart showing the basic principle of the algorithms used in Supplementary Fig. 8 with details in the figure caption.

For practical implementations, we chose the modal decomposition approach^{24,28} and the probe is expanded into several eigenmodes of the density matrix formed by a mixed state. The total intensity of all eigenmodes are normalized to the measured intensity of the diffraction patterns. A flowchart of the algorithms showing the basic principle is given in Supplementary Fig. 8.

The workflow of the algorithm is schematically shown in Supplementary Fig. 8 and more details are described as a ptychography toolkit in ref.⁷¹.

Supplementary Fig. 8. A flowchart of mixed-state electron ptychography. The diagram illustrates the working principle of the LSQ-ML ptychography method [49]. At first, initial guesses of the exit-waves computed from the illumination probe modes and complex-valued object function are forward-propagated to the reciprocal space, where they are updated in order to maximize a likelihood function given the measured data. The updated reciprocal models are then back-propagated to the object plane, where a linear least-squares method is used to determine the optimal decomposition into the incoherent probe modes, the shared object function and to refine the estimated scan positions. Finally, new estimates of the exit-waves are calculated from the updated probes and the object and forward-propagated. This procedure is iteratively repeated till convergence.

Some other points:

To study the different doses, it is stated that the current has been varied. It has to be discussed how this was done, and what impact it has on the coherence properties of the beam. Given that the brightness (current/phase space volume) along the beam line is a conserved quantity, coherence and current are hard to disentangle, e.g. increasing the probe demagnification by exciting the C1 lens (concomitantly decreasing the current) should improve coherence etc.... of course there are effective sources of decoherence like sample vibrations, chromatic aberrations etc - but this needs to be discussed.

We use a monochromator defocus to control the beam current. It works similarly as deactivating the C1 lens. We notice a slight increase of the probe coherence when reducing the current. But the change is tiny because the coherent current of the electron source is higher than 50 pA and the maximum current used is only 14 pA. We have added the discussions in the Methods section on page, 22, lines 16-19.

... via defocusing a monochromator. The coherence of the electron probe increases slightly when the beam current reduces but the change is not significant (< 3%) as the beam current used (0.1 - 14 pA) is always much lower than the coherent current of the source (~ 50 pA).

In a similar vein, is there a conjecture about how the shapes of the incoherent modes arise? Especially the third mode with its sixfold symmetry looks interesting.

There are many different origins of the probe incoherence which have been listed on page 19, lines 7-8. It is difficult to separate different contributions. Furthermore, the modal decomposition procedure gives orthogonal eigenmodes and different modes may not have direct physical meaning.

Is there an intuitive reason why many-mode reconstruction is less sensitive to the initial mode guess? It feels counterintuitive, given the introduction of another degree of freedom to a strongly non-convex problem.

We think the partial coherence of the probe can effectively be described via the mixed state model. It is more robust and thus can have a very good convergence even from a much worse initial guess. However, in the single mode reconstruction, the partial coherence of the probe is not modeled, and it is very unstable. Therefore, a much better initial guess is required. To clarify this point, we have added some statements in the Methods section on page 24, lines 5-9:

This seems counterintuitive but it is not surprising. Because mixed-state ptychography can account for the nonnegligible partial coherence of the probe and provide a more accurate reciprocal model, whereas single mode ptychography is not sufficient for modeling the probe incoherence and its convergence can become unstable if the initial probe deviates from the real probe significantly.

It is stated that an FRC analysis has been performed to estimate the resolution of the shown reconstruction. This requires more explanation. In single-particle analysis, an FRC between two single-particle reconstructions derived from data half-sets serves as an estimator for the spectral SNR. However, as no single-particle analysis is performed on identical particles in the images, it is not clear to me between which data sets FRC is calculated, and how a resolution criterion can be defined.

We used two phase images reconstructed from two separate datasets from the same scan region, which serves as two independent measurements. Practically, two datasets were selected from one single dataset in

every two diffractions at each dimension but with different starting points. We have also added some more details about the FRC analysis in the Methods section on page 24, lines 11-19.

Fourier ring correlation

For Fourier ring correlation (FRC)^{31,67}, we used two phase images reconstructed from two separate datasets from the same scan region, which serves as two independent measurements. Practically, two datasets were selected from one single dataset in every second scan step at each dimension but with different starting points. After ptychographic reconstruction, a global linear phase term due to the inherent ambiguities of ptychography⁶⁸ is removed by fitting as a 2D linear function. Two phase images are aligned using the sub-pixel precision cross-correlation algorithm⁶⁹. Before FRC analysis, the edges of the phase images were Apodized to avoid the artifacts introduced from boundary discontinuities³¹. The resolution is estimated by using the 1-bit threshold³¹.

The color map of figure 2d is inverted with respect to all others. If this is intentional, please state, why.

The diffractogram in Fig 2d is from a region containing monolayer, bilayer and not well-crystalline samples (Fig. 2a). The high frequency spots are more diffused. We tested different colors and it turns out the reversed color has the best contrast. As we do not intend to quantify the magnitude of Fourier coefficients but only the visibility and thus the reversed color does not affect the main conclusions.

I find it unsatisfactory that the software containing the key algorithm is only accessible on request instead of being immediately available on a public repository where it can be used and reviewed by other researchers, ideally under a standard open source license. What is the rationale behind this decision?

The main ptychography code is already publicly available at the PSI website. We have updated the code availability statement on page 25:

The codes developed at Cornell University will be freely available immediately after the manuscript is published via <https://github.com/muller-group-cornell>. The ptychography reconstruction toolkit developed at Paul Scherrer Institut, Switzerland, is available on the website via the link: <https://www.psi.ch/en/sls/csaxs/software>.

With these points addressed, I support publication of the paper in Nature Communications.

We greatly appreciate the reviewer's support for publication of the paper in Nature Communications.

Reviewers' comments:

Reviewer #2 (Remarks to the Author):

The revisions to this manuscript are mostly sufficient. In this second review, I have a few additional points concerning the implied (but false) novelty of partial coherence in electron ptychography (apologies to the authors for not catching this in my first review but I think it is important enough) and one further comment left over from my first review. Provided that the implied additional revisions are addressed with sufficient rigor, I would recommend publication.

1. Manuscript states "High coherent field emission guns are widely used as the electron sources in modern electron microscopes and so electron ptychography usually assumes only a pure coherent state of the illumination probe^{19,26,27}." And later the manuscript states "There are only a few proof-of-principle demonstrations of mixed-state electron ptychography^{28,29} and while these suggest the promise of the approach, to date no high-quality atomic resolution or sub-angstrom resolution reconstructions have been achieved."

The effects of partial coherence (mixed state) were explored in ref 18, and the effects were already well understood in electron ptychography, and more generally x-ray and electron phase retrieval, well before then. Ref 18 achieved a spatial resolution of about 1 Angstrom (this is "atomic resolution"). Ref 18 achieved reconstructions of high quality. One point of difference, however, is that in ref 18 partial coherence did not so dramatically affect the results, that is, assuming perfect coherence did not produce "completely wrong" reconstructions (nor should it). This is in contrast with the present manuscript where "completely wrong" reconstructions ARE seen - a result which is very surprising and not in itself comprehensible (as I wrote in my first review). In summary, the authors should revise the statements in their manuscript so that they more directly acknowledge the fact that partial coherence (or mixed state, different name for same thing) in electron ptychography is already well-known and well-understood from previous works.

2. Manuscript states "Similar setups with a defocused probe have been adopted previously^{17,18,26} to overcome the slow readout speed of conventional CCD cameras and limited stability of the imaging systems."

This statement is not correct. Yes, similar (essentially identical) setups HAVE been used but the probe was not defocused for the reasons stated by the authors, it was defocused as an essential part of the ptychographic scheme (and the authors have done the same thing).

3. First review: "A mixed-state model of the electron probe ...is able to account many of the different factors ... such as the finite electron source size, chromatic aberration, point spread function of the detector, sample vibration, and fast instabilities of the image-forming system" Should the detector PSF be included in this list? Finite source size (sum of incoherent probes) does reduce the interference between different diffraction paths, but the sharpness of the diffraction discs remains the same.

Authors' response: The detector PSF degrades the sharpness of the diffraction disc and reduces the coherence, details can be found in Ref. [23] and [24]. We have already included those references. The incoherence from the finite source size comes from the incoherent sum of diffractions from different spatial positions. Such incoherent summation results in reduced contrast of the coherent speckles due to the different phase change of the probe wave function induced by the sample. Therefore, finite source size can be naturally covered within mixed-state model. A good discussion has been given in Ref. [24].

The authors have not understood my point (and admittedly I did not write it very clearly). The detector PSF DOES blur the diffraction discs but the assumed mixed state model DOES NOT (it only reduces the interference between diffraction discs). So, the detector PSF produces effects which are not included by the mixed state model. So, the authors should revise their statement.

Please find the attached our revised manuscript. The edits of the texts in the main manuscript has also been labeled in red. The response to reviewers' comments is listed after each comment.

Reviewer #2 (Remarks to the Author):

The revisions to this manuscript are mostly sufficient. In this second review, I have a few additional points concerning the implied (but false) novelty of partial coherence in electron ptychography (apologies to the authors for not catching this in my first review but I think it is important enough) and one further comment left over from my first review. Provided that the implied additional revisions are addressed with sufficient rigor, I would recommend publication.

We appreciate the referee's additional comments and recommendations for publication in Nature Communications. The responses to the comments are listed as follows.

1. Manuscript states "High coherent field emission guns are widely used as the electron sources in modern electron microscopes and so electron ptychography usually assumes only a pure coherent state of the illumination probe^{19,26,27}." And later the manuscript states "There are only a few proof-of-principle demonstrations of mixed-state electron ptychography^{28,29} and while these suggest the promise of the approach, to date no high-quality atomic resolution or sub-angstrom resolution reconstructions have been achieved."

We address the proof-of principle comment later, where the reviewer discusses this in more detail – in short, we add Ref. [18] to this list. For now, we address the partial coherence comments.

The effects of partial coherence (mixed state) were explored in ref 18, and the effects were already well understood in electron ptychography, and more generally x-ray and electron phase retrieval, well before then.

We agree that the effects of the partial coherence have been studied in phase retrieval by x-ray and electrons. We had already cited related work in the x-ray and electron literature, such as Refs. [22-24], [26]. We have also added reference to earlier work in coherent diffractive imaging (prior to that in ptychography) and an additional reference about partial coherence, Ref. [25] on page 4, line 6,

As first demonstrated in coherent diffractive imaging²³ and X-ray ptychography²⁴, the reconstruction can be improved by introducing such state mixtures to account for the partial coherence of the probe^{22,25,26}.

25. Whitehead, L. W. *et al.* Diffractive imaging using partially coherent x rays. *Phys. Rev. Lett.* **103**, 243902 (2009).

We agree that Ref. [18] also considered the partial coherence in electron ptychography, although using a different approach, assuming a gaussian model for mutual incoherence, not the mixed state (more explicitly, modal decomposition) which is why it was not listed with the mixed-state approach. Ref. [18], however, also concluded that “We note that results without incorporating these corrections [i.e. partial coherence] into the reconstructions do not materially affect the results” so it would not be accurate to use this reference as an illustration of where including partial coherence improved resolution. Nevertheless, we still include Ref. [18] here but point out more explicitly the different approach adapted. (See pg 4, line 14 changes below)

Ref 18 achieved a spatial resolution of about 1 Angstrom (this is "atomic resolution"). Ref 18 achieved reconstructions of high quality. One point of difference, however, is that in ref 18 partial coherence did not so dramatically affect the results, that is, assuming perfect coherence did not produce "completely wrong" reconstructions (nor should it).

On page 3, line 14 in the introduction where we introduced Ref. [18], we had described it as “high-resolution”. We have added Ref. [18] to the list of earlier proof-of-concept experiments on page 4, line 15 (The reviewer had essentially questioned the omission from this list at the beginning of their comment). The questions of “high quality” is subjective so to make it clear we specify sub-Angstrom, a common benchmark for a modern tool. It is clear that the reconstruction in Ref. [18] neither achieved sub-angstrom resolution as is commonly reached by aberration corrected electron microscopes, nor surpassed the conventional resolving power of their microscope ($\sim 0.8 \text{ \AA}$ at 300 kV) as we do here. Therefore, it is suitable to describe it as proof-of-principle (page 4, line 15),

There are only a few proof-of-principle demonstrations of electron ptychography **considering the partial coherence via approaches either Gaussian blind deconvolution¹⁸ or modal decomposition^{29,30}** and while these suggest the promise of the approach, **to date no sub-angstrom resolution reconstructions have been achieved, even on instruments capable of sub-angstrom resolution in conventional imaging modes.**

One point of difference, however, is that in ref 18 partial coherence did not so dramatically affect the results, that is, assuming perfect coherence did not produce "completely wrong" reconstructions (nor should it). This is in contrast with the present manuscript where "completely wrong" reconstructions ARE seen - a result which is very surprising and not in itself comprehensible (as I wrote in my first review). In summary, the authors should revise the statements in their manuscript so that they more directly acknowledge the fact that partial coherence (or mixed state, different name for same thing) in electron ptychography is already well-known and well-understood from previous works.

The wrong reconstructions in this work only happen for very large scan step conditions, that is, a lack of constraints that occurs only when we push beyond the boundaries of conventional ptychography, e.g., for a faster acquisition (by reducing number of acquisitions) or large field-of-view (by reducing overlap). It is wrong to conclude that perfect coherence never works, nor did we claim that, in fact citing previous examples (e.g pg 11, line 22). We also demonstrated ‘high-quality’ reconstructions in a good probe overlapping conditions and assuming a perfect coherent probe, i.e., single-mode, in Fig. 3 and supplementary Fig. 4a & d, with an information transfer up to $\sim 1.1 \text{ \AA}^{-1}$. Single-state reconstructions are also shown in Figure 4 (previously a supp figure), both from our groups and others (Pelz, Song). Therefore, our results do not conflict with those statements in Ref. [18] nor do we imply it. We clarify the failure mode for our data in the main text on page 10, line 21:

artifact free reconstruction by single-state ptychography can only be achieved with a very small step size, 0.85 \AA , corresponding to a 95% probe overlap. **For larger scan step sizes, shown in Supplementary Fig. 4b-c, the single-state reconstructions fail to converge to the correct structure and generate artificial periodicities of the sample. At these large step sizes the errors in modeling of the probe partial coherence by the single-state reconstruction accumulate, leading to a solution that is not able to describe the measured data well and therefore, in combination with the weaker constrains due to larger scanning step, it leads to nonphysical reconstructions. Even in the case of a large probe overlap where the single-state reconstruction works, the mixed-state reconstruction has about two times better resolution and enhanced contrast compared to that from the single-state (Supplementary Fig. 4).**

2. *Manuscript states "Similar setups with a defocused probe have been adopted previously^{17,18,26} to*

overcome the slow readout speed of conventional CCD cameras and limited stability of the imaging systems."

This statement is not correct. Yes, similar (essentially identical) setups HAVE been used but the probe was not defocused for the reasons stated by the authors, it was defocused as an essential part of the ptychographic scheme (and the authors have done the same thing).

The reviewer's reading of that statement was not what we intended, and we agree the setup is being used as intended. There are 2 setups to ptychography – in focus, and defocused. The defocused approach has the benefit over the in-focus scheme of being able to address the above-mentioned problems, which is the point we were trying to make. To make this clear, we have readdressed the sentence (page 5, line 15) as,

Similar setups with a defocused probe have been adopted previously^{17,18,26}, **which has shown benefits for overcoming** the slow readout speed of conventional CCD cameras and limited stability of the imaging systems.

3. First review: "A mixed-state model of the electron probe ...is able to account many of the different factors ... such as the finite electron source size, chromatic aberration, point spread function of the detector, sample vibration, and fast instabilities of the image-forming system" Should the detector PSF be included in this list? Finite source size (sum of incoherent probes) does reduce the interference between different diffraction paths, but the sharpness of the diffraction discs remains the same.

Authors' response: The detector PSF degrades the sharpness of the diffraction disc and reduces the coherence, details can be found in Ref. [23] and [24]. We have already included those references. The incoherence from the finite source size comes from the incoherent sum of diffractions from different spatial positions. Such incoherent summation results in reduced contrast of the coherent speckles due to the different phase change of the probe wave function induced by the sample. Therefore, finite source size can be naturally covered within mixed-state model. A good discussion has been given in Ref. [24].

The authors have not understood my point (and admittedly I did not write it very clearly). The detector PSF DOES blur the diffraction discs but the assumed mixed state model DOES NOT (it only reduces the interference between diffraction discs). So, the detector PSF produces effects which are not included by the mixed state model. So, the authors should revise their statement.

We had simply included the discussion of detector PSF correction because it has been demonstrated in previous work – Ref. [22], Suppl Fig 3 shows a concrete example of a successful PSF correction. However, we also noted the Ref. [22] was missed unintentionally in the citations in this sentence (although it is cited elsewhere), so it is understandable how the reviewer may have missed it. Our view is this is not our fight. The PSF of our detector is very good and we do not consider it as a main source of the partial coherence. Given our good PSF, it has no effect on our results whether or not the algorithm corrects for the detector PSF. Accordingly, we have removed that claim. We retained Ref. [22] as it also covers the other discussed sources of decoherence that the reviewer agrees with. The sentence (page 19, line 10) has been revised as,

A mixed-state model is beneficial for ptychographic reconstructions using the data from current electron microscopes, as it is able to account for many of the different factors that result in decoherence-like effects

in measured diffraction patterns^{21,22,42,53}, such as the finite electron source size, chromatic aberration, sample vibration, and fast instabilities of the image-forming system.